# Spatio-Temporal Pattern and Spatial Disequilibrium of Cultivated Land Use Efficiency in China: An Empirical Study Based on 342 Prefecture-Level Cities

Yajuan Wang [1], Xi Wu [2] and Hongbo Zhu [1,*]

[1] Department of Land Resource and Real Estate Management, School of Public Administration, Sichuan University, Chengdu 610065, China
[2] School of Economics, Sichuan University, Chengdu 610065, China
* Correspondence: zhb@scu.edu.cn

**Abstract:** As an important resource for human survival and development, the utilization efficiency of cultivated land is directly related to national food security and social harmony and stability. Based on the stochastic frontier production function, this paper estimated the cultivated land use efficiency of 342 prefecture-level administrative regions in China from 2003 to 2019 and used spatial autocorrelation analysis and the Gini coefficient decomposition model to explore the spatial agglomeration and spatial disequilibrium of cultivated land use efficiency in China. The results showed the following: (1) Overall, the efficiency of cultivated land use in China has steadily improved since 2003, but the overall level remains low. The use efficiency of cultivated land decreases in the order of eastern, northeastern, western, and central regions, and the use efficiency of cultivated land in the central region increased the fastest. (2) From the perspective of the spatial dimension, the cultivated land use efficiency in urban areas of China has a multi-core structure of "high in the south and low in the north, high in the east and low in the west" and an obvious spatial differentiation pattern. At the same time, the spatial aggregation characteristics of cultivated land use efficiency have become more prominent with the passage of time. (3) There are obvious regional differences in cultivated land use efficiency in China, showing a downward trend as a whole, and the gap between regions is the main source of spatial non-equilibrium, followed by the super-variable density and the gap within regions. Revealing the temporal and spatial changes in cultivated land use efficiency is helpful to understand the present situation of cultivated land use and to formulating effective land use policies.

**Keywords:** land use efficiency; stochastic frontier analysis; spatio-temporal pattern; spatial disequilibrium





## 1. Introduction

As an important resource for human survival and development, the utilization efficiency of cultivated land is directly related to national food security and social harmony and stability, especially in the context of the COVID-19 epidemic and turbulent global political and economic patterns [1,2]. Cultivated land resources have a direct impact on food security and, in turn, on human survival and development [3]. Although many countries have developed a series of policies and regulations on cultivated land protection and food security, the conflict between population and cultivated land still exists [4–6]. Especially for China, the basic condition of the country is that there are many people and limited productive land, so the protection of cultivated land resources is even more important [7]. With the acceleration of urbanization since the reform and opening up, the phenomenon of cultivated land resources being occupied by the disorderly expansion of construction land has become common, and the continuous decline of the quantity and scale of cultivated land leads to a gradual reduction of the effective supply of cultivated land resources [8,9]. According to statistics, in 2019, China's cultivated land area decreased

by 154,300 square kilometers compared to 2018. Cultivated land pollution, fragmentation, and other practical problems all affect China's food security and ecological security, falling far short of the concept of sustainable economic and social development [10–12]. The protection of cultivated land resources is a matter of urgency. Studies have shown that there has been spatial heterogeneity and aggregation of cultivated land use efficiency in different periods [13,14]. Therefore, scientifically measuring the efficiency of cultivated land use, exploring the spatial and temporal changes and regional differences, and carrying out an in-depth analysis of the spatial aggregation and heterogeneity of cultivated land use efficiency, have become effective ways to comprehensively understand the efficiency of cultivated land resource use in China and can provide scientific reference for regional cultivated land use and conservation [15,16].

The construction of the evaluation system for cultivated land use efficiency by scholars came from the understanding of the connotation of cultivated land use efficiency [17], which mainly focuses on three aspects. First, from the perspective of a single index, it is measured by a single input or output of a unit area of cultivated land or unit labor force [18]. Second, the analysis framework of cultivated land use is constructed [19]. Some scholars select the appropriate indicators of cultivated land use efficiency and then use principal component analysis and other methods to calculate the cultivated land use efficiency [20,21]. Third, many scholars construct a land use model of the "input-output" system, using data envelopment analysis [22], stochastic frontier analysis [23,24], the SBM model [25], and the Malmquist production index [26]. Many scholars have used data envelopment analysis to evaluate the efficiency of cultivated land use. Wang et al. analyzed the cultivated land use efficiency in Southwest China from 2000–2015 by the DEA method [27]. The results showed that the high-value areas of pure technical efficiency and scale efficiency have been expanding over time. Among them, the comprehensive efficiency mainly reflects the relative scale of input and output of cultivated land use. The slack-based measure (SBM) model is often used when there are undesired outputs in the model. Kuang et al. analyzed the inter-provincial cultivated land use efficiency in China from 2000–2017 by incorporating carbon emissions into the analytical model of cultivated land use efficiency [28]. During the study period, the national cultivated land use efficiency showed an overall increasing trend, but there were still many provinces with low cultivated land use efficiency. It further corroborates the authenticity of this study. Ye et al. used a stochastic frontier production function to measure the spatial and temporal variation of cultivated land use efficiency in each province of China from 1999 to 2008 and constructed a land-average input-output panel data model [29]. The stochastic frontier production function is used to measure the efficiency of cultivated land use in terms of land-average input and output, which can effectively reflect the efficiency of input and output per unit of the cultivated land area [30]. Among them, the stochastic frontier analysis method can fully consider the influence of stochastic factors and technical inefficiency and has gradually become the mainstream tool for efficiency calculation [31]. There are two main levels of research on the spatial pattern of cultivated land use efficiency: one is the level of the spatial research scope, which is mostly based on the whole country [17], urban agglomeration [32], watershed [33], inter-provincial [28], and major grain-producing areas [13]. The second is the method of spatial pattern analysis. Most scholars conduct spatial analysis based on their research objects, using kernel density estimation [34], the spatial center of gravity model [35], spatial autocorrelation, and other research methods to explore the spatial and temporal distribution patterns of research objects at different levels [36]. However, a few scholars have studied the differences in cultivated land use efficiency at the regional level and brought the spatial imbalance into the spatial pattern, which plays an important role in the presentation of the spatial pattern of cultivated land use efficiency [37]. The Dagum Gini coefficient decomposition method can explore the source and composition of non-equilibrium on the basis of measuring the spatial differences and non-equilibrium of cultivated land use, and effectively solve the decomposition problem of regional differences and the sample description problem from individual to the region [38,39]. Obviously,

most of the above studies were based on efficiency measurement and further promoted by spatial exploration and mechanism analysis, but the existing studies have been less involved in the spatial distribution characteristics of cultivated land use efficiency at the national urban scale [40]. Meanwhile, DEA or SBM models are heavily used by cultivated land use efficiency measurement, which cannot fully consider the effects of stochastic factors and technical inefficiencies. To address these research gaps, we used the SFA model to measure the efficiency of cultivated land use in 342 prefecture-level administrative regions in China and analyzed their spatial aggregation and spatial differences. On the one hand, cultivated land use efficiency is determined by the input and output of cultivated land, while the input of cultivated land is affected by regional endowments and social and economic activities, and the input factors have certain spatial aggregation characteristics, so the cultivated land use efficiency may also have similar spatial characteristics [41]. On the other hand, the inter-regional and intra-regional differences in different regions may lead to the spatial differences and non-equilibrium characteristics of cultivated land use efficiency in China [42–44].

In view of the above problems, the purpose of this study is to analyze the changing trend and spatial distribution pattern of cultivated land use efficiency by measuring the value of cultivated land use efficiency in China and to contribute ideas for the coordinated development of regional cultivated land resources as well as cultivated land conservation. Specifically, we (1) measured the cultivated land use efficiency of 342 prefecture-level cities (autonomous prefectures, regions, leagues) in China using stochastic frontier analysis and analyzed the changes in regional cultivated land use efficiency from 2003 to 2019; (2) explored the spatial aggregation of cultivated land use efficiency using spatial autocorrelation analysis; and (3) explored the regional differences in cultivated land use efficiency and their sources using Dagum Gini coefficient analysis [45–47].

## 2. Materials and Methods

### 2.1. Materials

In this paper, when calculating the cultivated land use efficiency, the input indicators included the average input of primary industry labor force L (person/hectare), the average total power of agricultural machinery K (kilowatt/hectare), and the average amount of chemical fertilizer C (kg/hectare). The output indicator was the average total agricultural output value $Y$ (10,000 yuan/hectare). In order to eliminate the influence of price changes and ensure the comparability of data, we transformed the output value into constant prices using the base year 2003. The relevant data for the input and output indicators came from the China Regional Statistical Yearbook (2004–2014), the China Urban Statistical Yearbook (2004–2020), the statistical yearbooks of some provinces and cities, and national economic and social development bulletins.

### 2.2. Methods

#### 2.2.1. Stochastic Frontier Analysis

The green development and sustainable utilization of cultivated land resources should never rely on a large number of disorderly inputs of natural or economic factors but should focus on the improvement of factor utilization efficiency [48]. The input of cultivated land production factors and the utilization efficiency of agricultural production factors act on the output of cultivated land together. Generally, there are two methods to measure efficiency: the nonparametric method and the parametric method. Mathematical programming and data envelopment analysis are nonparametric methods. When the mathematical programming method is used to calculate the efficiency, the estimation of the boundary value is obtained from sub-samples, so it is particularly sensitive to outliers. Its disadvantage is that the influence of random error on individual efficiency is not considered [29]. The stochastic frontier analysis used in this paper belongs to the parametric method, which separates the inefficiency term from the random error term to ensure the efficiency of the estimated individual and considers the influence of random error on individual efficiency. Based

on this, this study used the stochastic frontier analysis method, according to the model of Battese and Coelli [49]. At the same time, in order to avoid the problem of collinearity, the Cobb–Douglas production function was used to construct the function model of cultivated land use efficiency in urban areas of China. The details are as follows:

$$Y_{it} = AL_{it}^{\beta_1} K_{it}^{\beta_2} C_{it}^{\beta_3} e^{v_{it} - u_{it}} \tag{1}$$

$$u_{it} = e^{-\eta(t-T)} u_i \tag{2}$$

$$TE_{it} = e^{-u_{it}} \tag{3}$$

$$\gamma = \frac{\sigma_u^2}{(\sigma_u^2 + \sigma_v^2)} \tag{4}$$

where $Y_{it}$ indicates the average agricultural output value of each city (10,000 yuan/ha). $L_{it}$ indicates the average number of primary sector laborers per land (person/ha), $K_{it}$ indicates the average total power of agricultural machinery per land in each municipality (kW/ha), and $C_{it}$ indicates the average fertilizer use per land in each municipality (kg/ha). $v_{it} \sim N(0, \sigma_v^2)$, subject to a normal distribution, represents random error terms that cannot be controlled by human beings, such as the impact of climate and disasters; $u_{it} \sim N^+(\mu, \sigma_u^2)$, subject to a non-negative one-sided normal distribution, represents the production inefficiency term for city $i$ in year $t$. $T$ is the time variable; $i$ and $t$ indicate the region and year, respectively. $TE_{it}$ indicates the cultivated land use efficiency of city $i$ in year $t$. $\beta_1$, $\beta_2$ and $\beta_3$ are the elasticity coefficients. In the calculation results, when $\gamma = 0$, it indicates that OLS can be directly used for measurement.

2.2.2. Spatial Autocorrelation Analysis

Spatial autocorrelation analysis is an effective means to analyze the spatial pattern, which can test whether a spatial distribution is aggregated. It includes global spatial autocorrelation analysis and local spatial autocorrelation analysis [50,51]. First, with the help of Global Moran's I, we carried out the spatial correlation study, and the calculation formula is as follows:

$$I = \frac{\sum_{i=1}^n \sum_{j \neq i}^n W_{ij} \left( F_i^t - \overline{F^t} \right) \left( F_j^t - \overline{F^t} \right)}{S^2 \sum_{i=1}^n \sum_{j \neq i}^n W_{ij}} \tag{5}$$

where, to measure the degree of spatial correlation of cultivated land use efficiency, the value $I$ is between [–1,1]; when $I$ is greater than 0, it has positive spatial correlation, when $I$ is less than 0, it has negative spatial correlation, and when $I$ takes 0, it indicates that there is no correlation. N is the number of space units; $F_i^t$ is the spatial score of the ith spatial unit at time t; $\overline{F^t}$ is the average value of cultivated land use efficiency; and $S^2$ is the variance of cultivated land use efficiency.

Subsequently, the Local Indicators of Spatial Association (LISA) was carried out to reveal the autocorrelation types of cultivated land use efficiency. According to the different quadrants of the study objects in the scatter diagram, the cultivated land use efficiency was divided into four spatial aggregation types: High-High Cluster (HH), High-Low Outlier (HL), Low-High Outlier (LH), and Low-Low Cluster (LL).

2.2.3. Regional Difference Decomposition Model

There are many ways to measure the regional difference, including the coefficient of variation, theil index, and the Gini coefficient. However, these methods have limitations in that they cannot further describe the characteristics of sub-regions and the sources of regional disparities. However, the Dagum–Gini coefficient method can solve the problem of decomposition and sample description in regional differences [52]. Thus, the Dagum–Gini coefficient was used to further analyze the differentiation law of cultivated land use

efficiency in the region. Firstly, the Gini coefficient reflecting the difference in cultivated land use efficiency was calculated as follows:

$$G = \frac{1}{2n^2\mu} \sum_{i=1}^{k} \sum_{j=1}^{k} \sum_{h=1}^{n_i} \sum_{r=1}^{n_j} |y_{ih} - y_{jr}| \tag{6}$$

where $G$ is the Gini coefficient, which measures the overall difference in cultivated land use efficiency in China; $y_{ih}(y_{jr})$ is the value of cultivated land use efficiency of $i(j)$ city in $h(r)$ sub-region; $\mu$ is the average value of cultivated land use efficiency of each city in the study area; $n$ is the total number of cities; $k$ is the number of divided regions; and $n_i(n_j)$ is the number of cities in $i(j)$ sub-region.

Secondly, the average value of urban cultivated land use efficiency in the sub-regions was sorted, $\mu_1 \leq \cdots \mu_i \leq \mu_j \leq \cdots \leq \mu_k$, and on this basis, the overall Gini coefficient was decomposed into three parts: $G_w$, $G_{nb}$, and $G_t$. The calculation formula is as follows:

$$G = G_w + G_{nb} + G_t \tag{7}$$

$$\begin{cases} G_w = \sum\limits_{i=1}^{k} m_i s_i G_{ii} \\ G_{ii} = \frac{1}{2n^2\mu} \sum\limits_{h=1}^{n_i} \sum_{r=1}^{n_j} |y_{ih} - y_{jr}| \end{cases} \tag{8}$$

$$\begin{cases} G_{nb} = \sum\limits_{i=2}^{k} \sum_{j=1}^{i-1} (m_j s_i + m_i s_j) G_{ij} D_{ij} \\ G_{ij} = \frac{1}{n_i n_j (\mu_i + \mu_j)} \sum\limits_{h=1}^{n_i} \sum_{r=1}^{n_j} |y_{ih} - y_{jr}| \end{cases} \tag{9}$$

$$G_t = \sum_{i=2}^{k} \sum_{j=1}^{i-1} (m_j s_i + m_i s_j) G_{ij} (1 - D_{ij}) \tag{10}$$

In the above formula, $G_{ii}$ represents the intra-regional Gini coefficient of the $i$ subregion; $G_{ij}$ represents the inter-regional Gini coefficient of $i$ and $j$ subintervals; $m_i = n_i/n$ is the share of the number of cities in the sub-region in the study area; $s_i = n_i\mu_i/(n\mu)$ is the share of the sum of cultivated land use efficiency in the sub-region in the study area; and $D_{ij}$ is the relative influence degree of cultivated land use efficiency between the $i$ and $j$ sub-regions. The calculation formula is as follows:

$$\begin{cases} D_{ij} = (d_{ij} - p_{ij})/(d_{ij} + p_{ij}) \\ d_{ij} = \int_0^\infty dF_i(y) \int_0^y (y - x) dF_j(x) \\ p_{ij} = \int_0^\infty dF_j(y) \int_0^y (y - x) dF_i(x) \end{cases} \tag{11}$$

In the formula, $d_{ij}$ represents the difference value of cultivated land use efficiency between the $i$ and $j$ sub-areas; that is, the mathematical expectation of the sum of sample values meeting $y_{ih} - y_{jr} > 0$ in the $i$ and $j$ sub-areas; $p_{ij}$ is the hypervariable first-order distance, which is the mathematical expectation of the sum of the sample values satisfying $y_{jr} - y_{ih} > 0$ in the sub-areas of $i$ and $j$; and $F_i(F_j)$ is the cumulative distribution function of the $i(j)$ subregion.

## 3. Results

### 3.1. Measurement of Cultivated Land Use Efficiency in Urban China

Using the computer program Frontier 4.1, the calculation of cultivated land use efficiency and the presentation of related parameters in China's urban areas from 2003 to 2019 were completed (Table 1). The model applicability test and parameter estimation test are two aspects of results verification in stochastic frontier analysis. Among them, the one-sided likelihood ratio test statistic LR = 1000.69 indicated the existence of random errors, so it was necessary to use the analysis method of the stochastic frontier production

function in this paper. $\gamma$ = 0.6266, and the generalized likelihood ratio statistical test is significant at the level of 1%, indicating the existence of random error terms, which further illustrated the need for model selection and the rationality of model construction in this paper. At the same time, all parameters have passed the significance test at the level of 1%, which shows that the model we built can well explain the cultivated land use efficiency in China.

**Table 1.** Stochastic frontier production function model's estimated value.

| Parameters to Be Estimated | Ln $A$ | $\beta_1$ | $\beta_2$ | $\beta_3$ | $\gamma$ | $\mu$ | $\eta$ |
|---|---|---|---|---|---|---|---|
| Coefficient | 1.5469 *** | 0.1033 *** | 0.4526 *** | 0.3209 *** | 0.6266 | 0.6889 | 0.2350 |
| Standard deviation | 0.0791 | 0.0258 | 0.0311 | 0.0282 | 0.0227 | 0.0526 | 0.0135 |
| T-statistic | 19.5601 | 4.0089 | 14.5667 | 11.3632 | 27.5645 | 13.0855 | 17.3816 |

Notes: *** Denote statistical significance at the 1% levels.

Based on the calculation results, it can be seen that the three input factors had a positive impact on the output value of cultivated land, and the output elasticity values of the three factors were 0.1033, 0.4526, and 0.3209, respectively. The output elasticity of the total power of agricultural machinery was the highest, followed by the amount of chemical fertilizer, and finally the number of employees in the primary industry. At present, with the continuous improvement of agricultural mechanization and the scale rate, the higher the degree of scale and mechanization, the greater the impact on the output value of cultivated land. However, for the labor input in the use of cultivated land, with the continuous advancement of urbanization, more and more potential laborers in rural areas are choosing to go out to work to improve their incomes and the rural labor force has decreased sharply, leading to the small elasticity of labor as an input factor. At the same time, the sowing and harvesting of crops is reliant more and more on mechanical and technological means, which is in line with the reality.

The value of cultivated land use efficiency in different regions had certain regional differences. According to the actual situation of social and economic development in China, this paper divided the research unit into four regions: the eastern, central, western, and northeastern regions. From the perspective of the regional evolution trend (Figure 1), the overall level of cultivated land use in China is low, and there is a significant gap in the efficiency of cultivated land use in different regions, but the trend of change over time is similar. The cultivated land use efficiency values are consistently higher than the average of China's cultivated land use efficiency in the eastern and northeastern regions, and consistently lower than the efficiency average in the western and central regions. Generally speaking, the increasing trend of cultivated land use efficiency in each region is relatively stable, with an average added value of 0.230. In terms of the rate of change, the growth rate in the central region is the highest, while that in the eastern region is the lowest. This indicates that there is a large gap between the actual output and the potential output of cultivated land in China at the present stage, with the gap gradually narrowing during the study period. The large-scale planting and higher planting level of cultivated land in the eastern region affected the level of cultivated land use efficiency, but there is still room for improvement in the overall efficiency level.

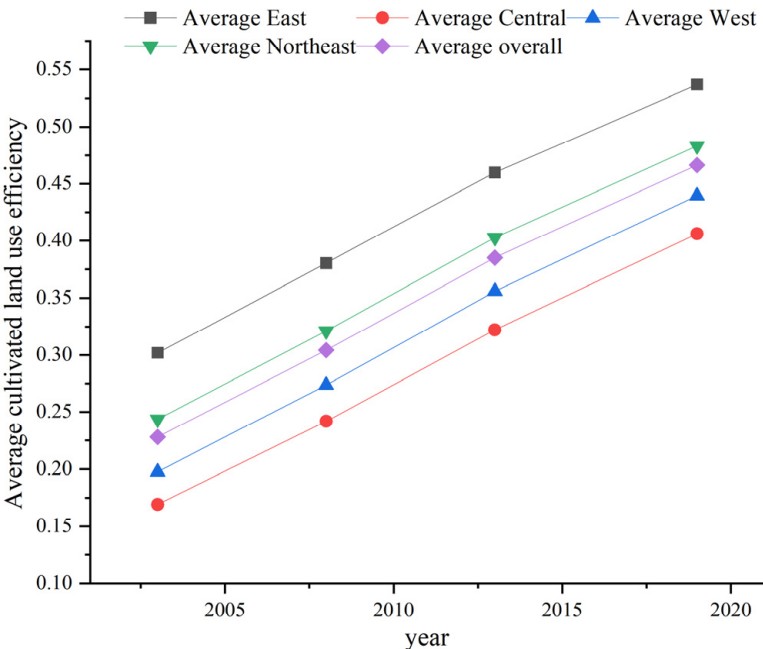

**Figure 1.** Trends of cultivated land use efficiency in China from 2003 to 2019.

*3.2. Spatio-Temporal Pattern Analysis of Cultivated Land Use Efficiency*

3.2.1. Analysis of Spatio-Temporal Characteristics of Cultivated Land Use Efficiency

In the same period, there were regional differences in the efficiency of cultivated land use in China, and the efficiency of cultivated land use has been improving over time (Figure 2). In 2003, the average value of cultivated land use efficiency in China was 0.222. The high-score areas were concentrated in the Yangtze River Delta, the Pearl River Delta, the Northeast Plain, and the Sichuan Basin. These areas have better hydrothermal climate conditions, however, the flat terrain of the plain areas is more conducive to large-scale cultivation of crops. A higher level of cultivated land use efficiency was determined by many factors. The low score areas were mostly distributed in the arid and semi-arid areas of the north and the Qinghai-Tibet Plateau, where the water and heat conditions are poor and the level of agricultural production technology is low. In 2008, the average value of cultivated land use efficiency in China was 0.298, an increase of 34.21% compared with the average value in 2003. The cultivated land use efficiency in the central region of China grew the fastest, reaching 42.85%. The spatial distribution pattern was basically consistent with that from 2003, and the change was mainly manifested in the general increase of the value of cultivated land use efficiency in various regions. In 2013, the average efficiency of cultivated land use in China was 0.379. In terms of the spatial pattern, the high score areas of cultivated land use efficiency in China are still concentrated in the southern coastal areas, the middle and lower reaches of the Yangtze River, and other areas. With their good natural endowments, the average value of cultivated land use efficiency in these areas is always higher than that in other areas. In 2019, the average value of cultivated land use efficiency in China was 0.461, and the growth rate of the average value of cultivated land use efficiency in China was 1.075 during the study period. The use efficiency of cultivated land in all regions increased, and the spatial distribution was relatively consistent. On the whole, from 2003 to 2019, there was an obvious spatial differentiation pattern of cultivated land use efficiency in China's urban areas, showing a multi-core concentrated distribution pattern of "high in the south and low in the north, high in the east and low in the west", and "high in the southwest and low in the northwest".

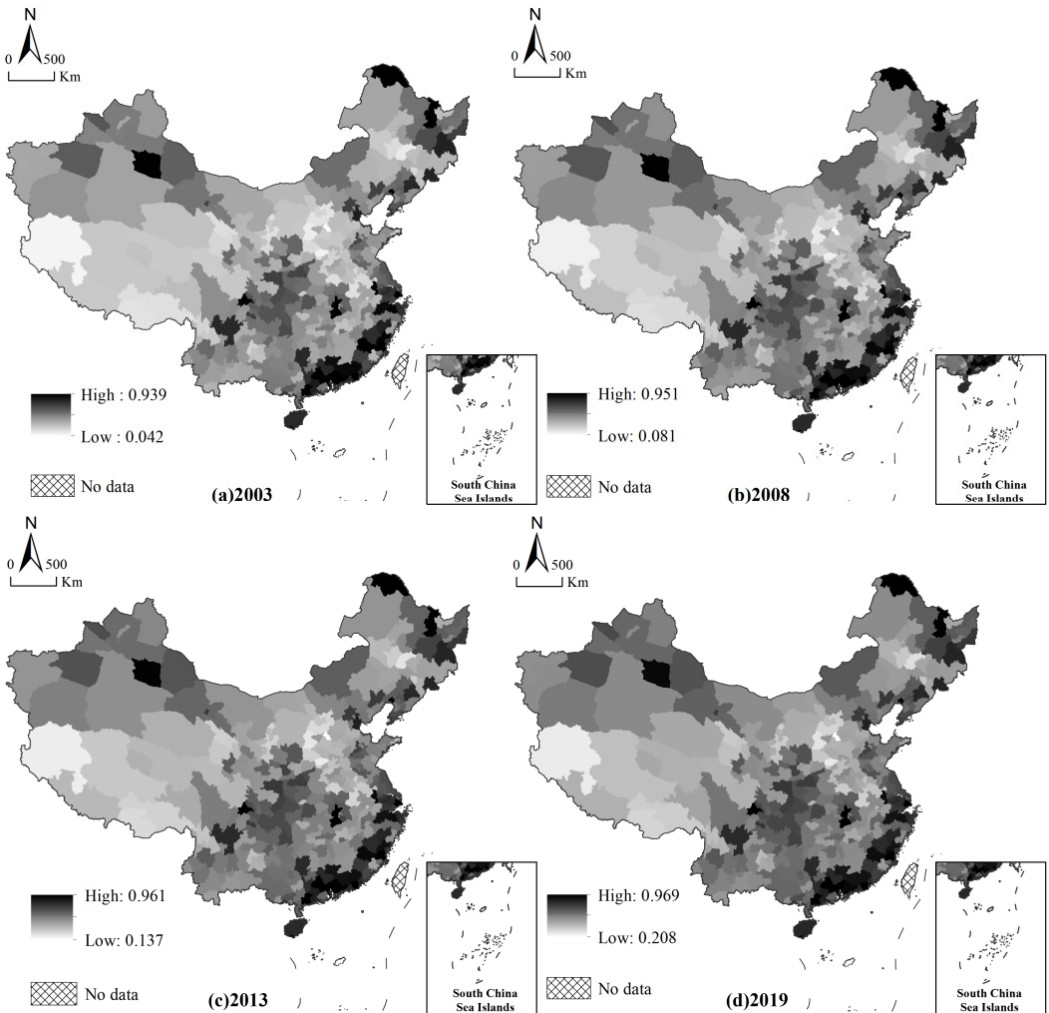

**Figure 2.** Results of cultivated land use efficiency from 2003 to 2019.

3.2.2. Analysis of Spatial Aggregation of Cultivated Land Use Efficiency

From 2003 to 2019, the Moran's I index of cultivated land use efficiency at the municipal level in China was significant at the level of 1%, with values of 0.427, 0.454, 0.469, and 0.479, respectively (Figure 3), indicating that the cultivated land use efficiency in China was positively aggregated in space, and the spatial clustering trend has become more pronounced over time. Furthermore, on the basis of global spatial autocorrelation, the local spatial Moran value of cultivated land use efficiency was calculated by using the formula of the Local Moran's I index and is presented in the figure.

(1) HH efficiency region: During the study period, the HH-type regions of cultivated land use efficiency in China were mainly concentrated in the Yangtze River Delta and the southeast coastal areas. The quality of cultivated land in these areas was high, and the good climatic conditions and topography were more conducive to mechanized production. They are also traditional agricultural production areas in China. With the passage of time, the number of HH regions increased from 25 in 2003 to 31 in 2019, and most of the added cities are located in the Pearl River Delta region. This shows that the high-value area of cultivated land use efficiency in China has had an obvious spatial diffusion effect, which can influence and drive the cultivated land in the surrounding areas to develop in the direction of efficient use. (2) LL efficiency region: From 2003 to 2019, the LL-type regions of cultivated land use efficiency in China's urban areas were mainly concentrated in Inner Mongolia, Gansu, Tibet, and other northwest regions, and part of the central region. These regions generally have high cultivated land altitudes, poor transportation accessibility, a relatively low level of agricultural science and technology, and mainly grow non-grain

crops, with low-quality of cultivated land. This has led to a large gap between the input of cultivated land and the expected output. By 2019, there was no significant increase in the LL-type region. (3) LH and HL efficiency regions: The number of LH and HL efficiency regions of cultivated land use in urban areas of China from 2003 to 2019 was relatively small, showing a scattered distribution trend, mainly located around the high- or low-value regions of cultivated land use efficiency in the western and southern regions. From 2003 to 2019, the change in the local spatial aggregation distribution pattern of cultivated land use efficiency was relatively stable, and the replacement of the aggregation interval basically remained stable.

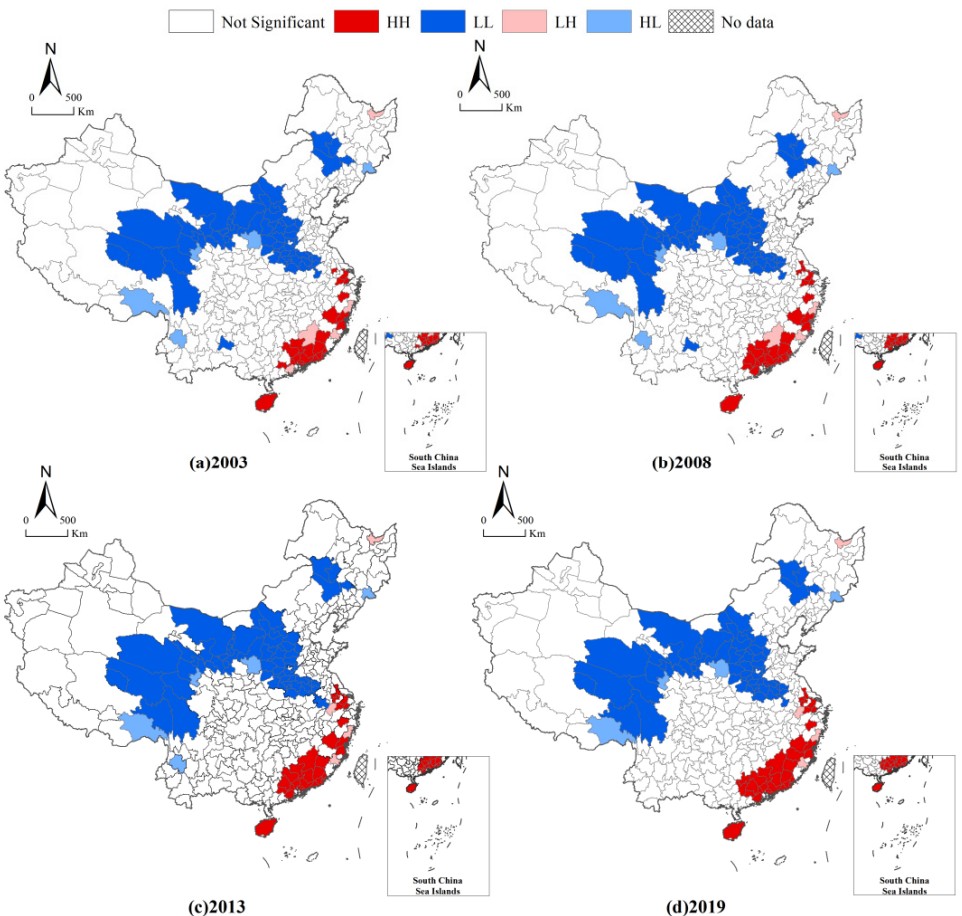

**Figure 3.** Spatial autocorrelation analysis results of cultivated land use efficiency in China from 2003 to 2019.

### 3.3. Analysis of the Spatial Non-Equilibrium of Cultivated Land Use Efficiency

3.3.1. Spatial Non-Equilibrium Characteristics of Cultivated Land Use Efficiency

There are obvious spatial differentiation characteristics of cultivated land use efficiency at the municipal scale in China (Table 2). Based on the calculation results of the national overall Gini coefficient, it can be seen that the overall Gini coefficient showed a continuous downward trend during the study period, from 0.2598 in 2003 to 0.1294 in 2019, and the spatial imbalance of cultivated land use efficiency in China gradually narrowed. The Gini coefficient of the eastern, central, western, and northeastern regions declined during the study period, which is consistent with the overall trend of the national Gini coefficient. During the study period, the eastern, northeastern, central, and western regions were arranged in descending order of spatial disequilibrium, and the spatial disequilibrium of cultivated land use efficiency in the eastern region was the largest, with an average Gini coefficient of 0.1855, while the spatial difference in the western region was the smallest, with an average Gini coefficient of 0.151. The range of the eastern and western regions

was 0.1266 and 0.1009, and the fluctuation range reflected the change degree of regional spatial non-equilibrium of cultivated land use in different regions.

**Table 2.** Gini coefficient and contribution rate of cultivated land use efficiency in China.

| Year | | 2003 | 2008 | 2013 | 2019 |
|------|------|------|------|------|------|
| G | China | 0.2598 | 0.2062 | 0.1634 | 0.1294 |
| $G_w$ | East | 0.2534 | 0.2016 | 0.1600 | 0.1268 |
| | Central | 0.2179 | 0.1736 | 0.1380 | 0.1097 |
| | West | 0.2050 | 0.1640 | 0.1309 | 0.1041 |
| | Northeast | 0.2476 | 0.1988 | 0.1589 | 0.1266 |
| $G_{nb}$ | East-Central | 0.3342 | 0.2661 | 0.2111 | 0.1671 |
| | East-West | 0.2857 | 0.2266 | 0.1794 | 0.1419 |
| | East-Northeast | 0.2640 | 0.2108 | 0.1679 | 0.1334 |
| | Central-West | 0.2221 | 0.1773 | 0.1412 | 0.1123 |
| | Central-Northeast | 0.2794 | 0.2227 | 0.1770 | 0.1404 |
| | West-Northeast | 0.2435 | 0.1944 | 0.1547 | 0.1229 |
| Contribution rate | $G_{nb}$ | 0.4708 | 0.4612 | 0.4540 | 0.4486 |
| | $G_t$ | 0.2855 | 0.2933 | 0.2991 | 0.3034 |
| | $G_w$ | 0.2437 | 0.2455 | 0.2469 | 0.2480 |

Based on the inter-regional Gini coefficient, the average Gini coefficient of the eastern and central regions was 0.2446, and the spatial disequilibrium degree was the largest. The average values of the east-west and central-northeast between [0.200 and 0.244] were 0.2084 and 0.2049, respectively, and the degree of spatial disequilibrium was moderate. East-northeast, west-northeast, and west-central had a smaller spatial disequilibrium. In terms of change trend, the non-equilibrium changes in cultivated land use efficiency in six regional combinations showed a significant downward trend, and the decline between different regional combinations was very close, with an average decline of 49.75%. The results showed that the inter-regional synergy of cultivated land use efficiency in China is gradually increasing, mainly manifested in the weakening of the degree of inter-regional disequilibrium at the spatial level due to the rapid increase of the value of cultivated land use efficiency in the central and western regions.

### 3.3.2. Unbalanced Sources of Cultivated Land Use Efficiency

Based on the source of spatial disequilibrium (Figure 4), during the study period, the three contribution rates were relatively stable over time and all had a certain effect on the spatial disequilibrium of cultivated land use efficiency. The average contribution rate of inter-regional disparity was as high as 45.87%, which is the main source of spatial non-equilibrium of cultivated land use efficiency, while the contribution rates of super-variable density and intra-regional disparity were similar at 29.53% and 24.60%, respectively. Thus, alleviating the spatial imbalance of cultivated land use efficiency at the municipal scale in China should focus on narrowing the regional disparity; that is, improving the cultivated land use efficiency in the central and western regions, and narrowing the regional disparity between the eastern and central regions. From the perspective of the change trend, from 2003 to 2019, the proportion of intra-regional differences and super-variable density in the overall contribution rate of cultivated land use efficiency in China gradually increased by 0.43% and 1.79%, respectively, while the inter-regional differences decreased by 2.22% during the study period.

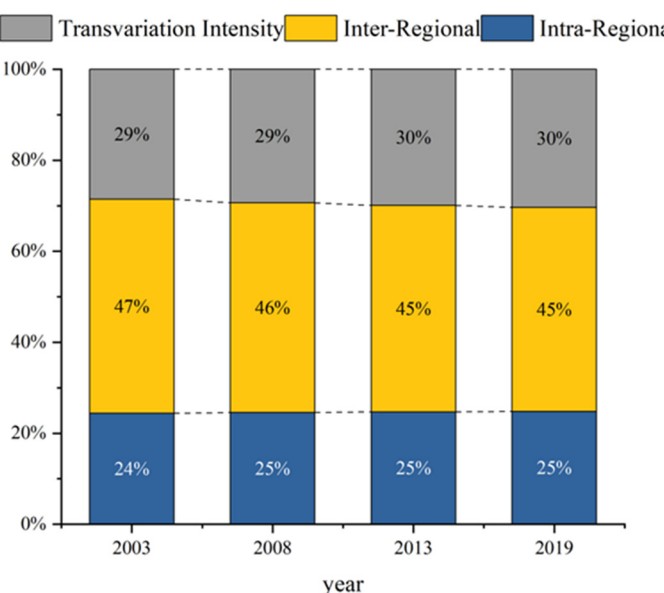

**Figure 4.** Spatial disequilibrium degree and source contribution rate.

## 4. Discussion and Conclusions

### 4.1. Discussion

The scarcity of resources and the finiteness of land determine that the improvement of the efficiency of cultivated land use is in the direction of sustainable development of cultivated land [53]. The prerequisite for improving the efficiency of cultivated land use is to understand the current situation and development trend of cultivated land use efficiency. Therefore, the main purpose of this study is to analyze the changing trend and spatial evolution pattern of cultivated land use efficiency by measuring the value of cultivated land use efficiency in China from 2003 to 2019, so as to provide the basis for sustainable utilization of cultivated land.

This study uses stochastic frontier analysis to carry out the measurement of cultivated land use efficiency in China and analyzes the change characteristics of cultivated land use efficiency in both time and space dimensions. Unlike other models for measuring efficiency, stochastic frontier analysis can separate the random error term from the inefficiency term and can fully consider the impact of random error on individual efficiency [49]. The application of this method in the measurement of cultivated land use efficiency has been recognized by many scholars [29]. Based on the results of the measurement of cultivated land use efficiency, it is clear that labor, capital, and technology inputs all have a positive effect on the improvement of cultivated land use efficiency. Among them, the output elasticity coefficient of total agricultural machinery power is the largest, which is 0.4526, indicating that the improvement of agricultural output in China at this stage mainly depends on the improvement of mechanization. This is consistent with Wang's conclusion [30]. The level of cultivated land utilization in China gradually increased during the study period, but remained low overall, indicating that the gap between the actual and potential output of cultivated land under established inputs is gradually narrowing.

This study further explored the spatial pattern of cultivated land use efficiency in terms of both spatial aggregation and spatial variation. The positive aggregation of cultivated land use efficiency in China has become more pronounced over time. That is, in most cases, high-value areas of cultivated land use efficiency tend to be adjacent to other areas of high levels of cultivated land use [54]. Therefore, this study should focus on the "spatial linkage" of cultivated land use efficiency, deepening the spatial diffusion effect, reducing the spatial polarization effect, and strictly preventing and controlling cultivated land in the areas with high efficiency of cultivated land utilization [55].To prevent the polarization of the non-grain phenomenon in high-efficiency areas, we should strengthen the supervision,

development, and protection of high-efficiency areas, and realize the key development of high-efficiency areas and the ecological protection of low-efficiency areas [56].

The calculation of the Dagum–Gini coefficient further illustrates the existence of spatial imbalances in the efficiency of cultivated land use. The regional difference in cultivated land utilization efficiency reflects the difference between input and output in the process of cultivated land protection and utilization, and it is also a difference in resource allocation [57]. This study found that inter-regional differences are the most important source of imbalance in the efficiency of cultivated land use. Therefore, an effective way to reduce the degree of spatial non-equilibrium would be by narrowing the gap in cultivated land use among regions and strengthening the sharing and association of agricultural resources and agricultural technologies in different regions, especially the coordination and development between adjacent cities. At the same time, by analyzing the changes in the sources of regional differences, on the one hand, the contribution rate of regional differences was dominant during the study period, the decline was small, and the impact of regional differences on the degree of disequilibrium of cultivated land use efficiency cannot be ignored. On the other hand, while focusing on the gap between regions, we also need to pay attention to the impact of the degree of crisscrossing between prefecture-level administrative regions on the efficiency of cultivated land use.

Further, referring to the existing research, some scholars have carried out the zoning of cultivated land, according to the results of spatial aggregation [58]. Xiong et al. divided HH-type regions with spatial agglomeration into key development areas. These areas usually have good natural and economic conditions and good productivity of cultivated land. Therefore, combining the calculation results of spatial aggregation and disequilibrium and the types of agricultural divisions in China, we can carry out similar zoning. For example, HH-type areas can be divided into HH Plain Key Development Zones, which can be used as a national key grain production base and bear important grain production responsibilities. For LL-type regions, it is necessary to further analyze the factors that affect the utilization efficiency of cultivated land, explore ways to improve the utilization efficiency of cultivated land and ensure the sustainable utilization of cultivated land resources.

### *4.2. Conclusions*

Based on the stochastic frontier production function model, this paper systematically measured the cultivated land use efficiency of 342 prefecture-level administrative regions in China from 2003 to 2019. The spatial autocorrelation analysis and Dagum–Gini coefficient analysis methods were used to depict the spatial agglomeration and spatial non-equilibrium distribution of cultivated land use efficiency in China. The main conclusions are as follows:

First, the output elasticity of the total power of agricultural machinery was the highest, followed by the amount of chemical fertilizer used, and finally the labor input, at 0.4526, 0.3209, and 0.1033, respectively. During the study period, the overall cultivated land use efficiency in China's urban areas showed an upward trend, but the value of cultivated land use efficiency was still generally low, with an average of 0.3398. There was an obvious spatial differentiation pattern, showing multi-core and concentrated distribution characteristics of "high in the south and low in the north, high in the east and low in the west". At the city level, the cities with higher cultivated land use efficiency were mainly concentrated in the Yangtze River Delta, Pearl River Delta, Northeast Plain, and Sichuan Basin, while the cities with lower cultivated land use efficiency were mostly distributed in the arid and semi-arid areas of the north and the Qinghai-Tibet Plateau, where the water and heat conditions are poor and the level of agricultural production technology is low. At the regional level, the trends of cultivated land use efficiency over time in the eastern, central, western, and northeastern regions were similar. During the study period, the eastern and northeastern regions were always higher than the average value of cultivated land use efficiency, while the central and western regions were slightly lower than the average value, which further illustrated the obvious regional characteristics of cultivated land use efficiency.

Second, the spatial agglomeration characteristics of cultivated land use efficiency at the municipal scale in China are significant, and the spatial agglomeration trend has become more prominent with the passage of time. During the study period, the HH-type regions were mainly distributed in the southeast coastal cities and the Yangtze River Delta region and had obvious spatial diffusion effects over time. LL-type is mainly concentrated in the northern and western regions of China, where the development of cultivated land resources is insufficient, the natural and hydrothermal conditions are relatively backward, the rugged and mountainous farming environment is not conducive to mechanized and large-scale production, and the utilization efficiency of cultivated land is therefore low. The number of LH and HL regions is small, and they are scattered in the western and southern regions of China, which are easily affected by the surrounding cultivated land use patterns and utilization status.

Third, the degree of spatial disequilibrium of cultivated land use efficiency in China generally showed a downward trend. At the level of intra-regional disparity, the Gini coefficient of each region had a similar downward trend, and the spatial imbalance within the region improved. At the level of inter-regional disparity, the eastern-central region, and the west-central region were the regions with the largest and smallest spatial disequilibrium, respectively. At the contribution rate level, the contribution degree of inter-regional disparity, super-variable density, and intra-regional disparity to the spatial non-equilibrium of cultivated land use efficiency in China decreased in turn. With the passage of time, the influence of super-variable density and intra-regional differences on the overall situation is increasing.

### 4.3. Limitations and Further Research

This study had certain limitations. When we constructed the utilization efficiency measurement model, we paid less attention to the ecological efficiency of cultivated land utilization. This may lead to inaccurate efficiency calculation results. However, according to the relevant literature studies, it can be seen that the spatial and temporal characteristics and change trends of the final results are consistent, which are in line with the research objectives of this paper [7]. Meanwhile, this study focuses more on the spatial layout and temporal evolution of cultivated land use efficiency in China, and less on the study of factors influencing cultivated land use efficiency. Therefore, in the next research, we will take the ecological efficiency of cultivated land into account, and at the same time explore the factors that affect the utilization efficiency of cultivated land, so as to contribute to the protection of cultivated land and food security.

**Author Contributions:** Conceptualization, H.Z. and Y.W.; methodology, H.Z. and X.W.; software, Y.W.; validation, H.Z. and X.W.; formal analysis, H.Z. and Y.W.; investigation, H.Z. and Y.W.; resources, H.Z. and Y.W.; data curation, X.W.; writing—original draft preparation, H.Z. and Y.W.; writing—review and editing, H.Z., Y.W. and X.W.; visualization, X.W.; supervision, H.Z. and Y.W.; project administration, H.Z. and Y.W.; funding acquisition, H.Z. All authors have read and agreed to the published version of the manuscript.

**Funding:** This research was funded by the National Social Science Fund (13BGL149) and the Statistical Development Special Project of the Sichuan Social Science Fund (SC19TJ024).

**Institutional Review Board Statement:** Not applicable.

**Informed Consent Statement:** Not applicable.

**Data Availability Statement:** Publicly available datasets were analyzed in this study. These data can be found at: https://data.stats.gov.cn/, and https://data.cnki.net/Yearbook/Navi?type=type&code=A, accessed on 25 May 2022.

**Acknowledgments:** The authors are particularly grateful to the anonymous reviewers for their comments and suggestions which contributed to the further improvement of this paper.

**Conflicts of Interest:** The authors declare no conflict of interest.

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
