# Peer review of "Spatio-Temporal Pattern and Spatial Disequilibrium of Cultivated Land Use Efficiency in China: An Empirical Study Based on 342 Prefecture-Level Cities"

_land, doi:10.3390/land11101763_

Round 1

Reviewer 1 Report

The document is fine, although I miss the baseline data for the statistical analysis.

I don't know if there is an evolution of cultivated land over time in the different areas. I think that a table showing the evolution of the cultivated area would help to understand the context. Is it possible to grow less but more efficiently?

Could it be that it is better cultivated in areas with greater economic possibilities?

I think figure 2 would look better in a color scale, but you don't need to modify it, it's just an opinion.

Author Response

The document is fine, although I miss the baseline data for the statistical analysis.

Q1: I don't know if there is an evolution of cultivated land over time in the different areas. I think that a table showing the evolution of the cultivated area would help to understand the context. Is it possible to grow less but more efficiently?

R1: Thanks for this comment. First, there is an evolution of cultivated land over time in the different areas. The most visual change in cultivated land change is the evolution of the cultivated area. Also, the parameters related to the efficiency of cultivated land use are the total power of agricultural machinery and the change in the intensity of fertilizer and pesticide use, etc. From 2003-2019, all these indicators have changed. Even for the same piece of cultivated land, the cultivated land use efficiency change due to the changes in technological inputs, capital inputs, and fertilizer and pesticide inputs[1]. Second, we have added sentences reflecting changes in cultivated land area, which can better reflect the context of the study. Finally, It is possible to grow less but more efficiently.The main reason is that the efficiency of cultivated land use is not determined only by the area of cultivated land, but also by the technical inputs, labor inputs, and capital inputs on the cultivated land[2]. The input of cultivated land production factors and the utilization efficiency of agricultural production factors act on the output of cultivated land together.

  1. Xie, H.L.; Zhang, Y.W.; Choi, Y. Measuring the Cultivated Land Use Efficiency of the Main Grain-Producing Areas in China under the Constraints of Carbon Emissions and Agricultural Nonpoint Source Pollution. Sustainability 2018, 10, doi:10.3390/su10061932.
  2. Kuang, B.; Lu, X.H.; Zhou, M.; Chen, D.L. Provincial cultivated land use efficiency in China: Empirical analysis based on the SBM-DEA model with carbon emissions considered. Technological Forecasting and Social Change 2020, 151, doi:10.1016/j.techfore.2019.119874.

Q2: Could it be that it is better cultivated in areas with greater economic possibilities?

R2: Thanks for this comment. We have added some sentences to illustrate them. From the results, it is true that the more economically developed regions generally have higher cultivated land use efficiency. This has also been confirmed by scholarly research[3]. We believe that there are two main reasons for this phenomenon. First, for China, there is a large gap between the natural conditions of cultivated land in the eastern and western regions. The topography of the eastern region is mostly plain, with concentrated and continuous cultivated land and a higher degree of intensification; the topography of the western region is mostly mountainous and hilly, with a higher degree of fragmentation of cultivated land and poorer natural conditions. Secondly, the economic development level in the eastern region of China is generally higher than that in the western region. Therefore, the capital input and agricultural technology input of cultivated land in the eastern region are also generally higher than those in the western region. Thus, although our study does not directly confirm the correlation between cultivated land use efficiency and economic development, this phenomenon exists.

  1. Han, H.B.; Zhang, X.Y. Exploring environmental efficiency and total factor productivity of cultivated land use in China. Science of the Total Environment 2020, 726, doi:10.1016/j.scitotenv.2020.138434.

Q3: I think figure 2 would look better in a color scale, but you don't need to modify it, it's just an opinion.

R3: Thanks for this comment. In Figure 2 our data are raster data, therefore, the color is gradient. Thanks again for your suggestion, in this figure the spatial variation of the arable land use efficiency can be visualized, therefore no changes were made to this figure.

Thanks a lot for your detailed and professional comments, it is very effective in improving the quality of the paper.

Reviewer 2 Report

Your paper tries to assess the spatial-temporal pattern and the spatial disequilibrium of cultivated land use efficiency in China from 2003 to 2019 based on data from prefecture-level cities.

In general, your study is well developed, but some fundamental questions are related to data, methods, and interpretation.

(1) "The output indicator was the average total output value Y (10,000 yuan/hectare)" (L107-108). Since the output values are measured in monetary units, it needs to be clarified and clearly stated whether the monetary units are in nominal (current) or real (constant) values. Please indicate clearly the measurement used.

(2) Your efficiency estimation does not cover the negative impacts of agricultural production on the environment, the so-called "eco-efficiency".  Thus, your estimation might be biased. You might draw the wrong conclusion. Thus, your approach needs to be better introduced and discussed in relation to the eco-efficiency literature.

(3) You claim that you "want to put forward relevant policy recommendations" (L91-92). However, since you can't explain the level of inefficiency in your model, it is very difficult if not impossible to draw policy recommendations based on your results. You can't base the policy recommendation in the discussion (L326-373) on your results. For instance, you recommend "improve the efficiency of cultivated land utilization by actively transferring cultivated land to new agricultural subjects such as skilled farmers, large gain farmers, and agricultural enterprises" (L 342-345). This recommendation is not supported by your results in any way. You also don't cite literature to support this recommendation. 

(4) The discussion needs to put the results in relation to the existing literature but also needs to critically reflect on the methods used and the validity of the results. Any shortcomings of the research should be mentioned.

Specific comments

 L49-52: Measuring efficiency is insufficient to "improve the efficiency of cultivated land resource use"! It is a necessary, but not sufficient condition. Factors affecting the efficiency level need to be identified. And those factors need to be manageable.

L54-92: Efficiency concepts (technical, allocative, scale efficiency) and eco-efficiency in relation to different measurement tools (DEA, SFA, SBM, etc.) need to be introduced more systematically.  

L99-100: "...explored a reasonable scheme for the spatial use and protection of cultivated land..."  Based on which methodology? Based on which kind of reasoning?

Figure 1: This uniform increase in efficiency makes me extremely skeptical. How could this be explained? I guess all values are average values over the regions and overall. Thus, the labels should be improved. "East" to "Average East" and so on and "Average" to "Average overall"  

Author Response

Your paper tries to assess the spatial-temporal pattern and the spatial disequilibrium of cultivated land use efficiency in China from 2003 to 2019 based on data from prefecture-level cities.

In general, your study is well developed, but some fundamental questions are related to data, methods, and interpretation.

Q1: "The output indicator was the average total output value Y (10,000 yuan/hectare)" (L107-108). Since the output values are measured in monetary units, it needs to be clarified and clearly stated whether the monetary units are in nominal (current) or real (constant) values. Please indicate clearly the measurement used.

R1: Thanks for this comment. We have added some sentences to illustrate this. This indicator is used in this study in constant prices based on 2003.

Q2: Your efficiency estimation does not cover the negative impacts of agricultural production on the environment, the so-called "eco-efficiency".  Thus, your estimation might be biased. You might draw the wrong conclusion. Thus, your approach needs to be better introduced and discussed in relation to the eco-efficiency literature.

R2: Thanks for this comment. We have added the reasons for choosing research methods. As for the ecological efficiency of cultivated land use, we realized that the lack of selection of unexpected output might lead to errors in the final result, so we added it to the discussion.

Q3:  You claim that you "want to put forward relevant policy recommendations" (L91-92). However, since you can't explain the level of inefficiency in your model, it is very difficult if not impossible to draw policy recommendations based on your results. You can't base the policy recommendation in the discussion (L326-373) on your results. For instance, you recommend "improve the efficiency of cultivated land utilization by actively transferring cultivated land to new agricultural subjects such as skilled farmers, large gain farmers, and agricultural enterprises" (L 342-345). This recommendation is not supported by your results in any way. You also don't cite literature to support this recommendation.

R3: Thanks for this comment. We have improved the discussion section by combining the results of the study and the literature. On the one hand, The positive aggregation of cultivated land use efficiency in China has become more pronounced over time. That is, high value areas of cultivated land use efficiency tend to be ad-jacent to other areas of high levels of cultivated land use[1]. Therefore,This should focus on the "spatial linkage" of cultivated land use efficiency, deepening the spatial diffusion effect, reducing the spatial polarization effect, and strictly preventing and controlling cultivated land in the areas with high efficiency of cultivated land utilization. On the other hand, inter-regional differences are the most important source of imbalance in the efficiency of cultivated land use. Therefore, an effective way to reduce the degree of spatial non-equilibrium would be by narrowing the gap of cultivated land use among regions and strengthening the sharing and association of agricultural resources and agricultural technologies in different regions, especially the coordination and development between adjacent cities[2]. Finally, we combined existing studies to partition the results of the measurement of cultivated land use efficiency[3]. For example, HH-type areas can be divided into HH Plain Key Development Zone, which can be used as a national key grain production base and bear important grain production responsibilities.

  1. Wang, Y.; Rao, Y.; Zhu, H. Revealing the Impact of Protected Areas on Land Cover Volatility in China. Land 2022, 11, doi:10.3390/land11081361.
  2. Zang, J.M.; Tang, C.Y.; Wang, Q.X.; Li, K.; LI, L.F. Research on Spatial Imbalance and Influencing Factors of Cultivated Land Use Efficiency in Guangdong Province Based on Super-SBM Model. China Land Science 2021, 35, 64-74, doi:10.11994/zgtdkx.20211008.085358.
  3. Xiong, C.S.; Zhang, Y.L.; Wang, Y.J.; Luan, Q.L.; Liu, X. Multi-function Evaluation and Zoning Control of Cultivated Land in China. China Land Science 2021, 35, 104-114, doi:10.11994/zgtdkx.20210916.155106.

Q4: The discussion needs to put the results in relation to the existing literature but also needs to critically reflect on the methods used and the validity of the results. Any shortcomings of the research should be mentioned.

R4: Thanks for this comment. We have improved the discussion and added a "Limitations and further research" section.

Specific comments 

Q5: L49-52: Measuring efficiency is insufficient to "improve the efficiency of cultivated land resource use"! It is a necessary, but not sufficient condition. Factors affecting the efficiency level need to be identified. And those factors need to be manageable.

R5: Thanks for this comment. We have recognized the error in the formulation of this paragraph and have corrected it.

Q6: L54-92: Efficiency concepts (technical, allocative, scale efficiency) and eco-efficiency in relation to different measurement tools (DEA, SFA, SBM, etc.) need to be introduced more systematically.

R6: Thanks for this comment. We have added more sentences to illustrate the different measurement tools.

Q7: L99-100: "...explored a reasonable scheme for the spatial use and protection of cultivated land..."  Based on which methodology? Based on which kind of reasoning?

R7: Thanks for this comment. We have revised this part and added more analysis in the discussion.

Q8: Figure 1: This uniform increase in efficiency makes me extremely skeptical. How could this be explained? I guess all values are average values over the regions and overall. Thus, the labels should be improved. "East" to "Average East" and so on and "Average" to "Average overall" 

R8: Thanks for this comment. Your comment made us aware of the errors in the presentation, here are all the averages for different areas, thank you for your comment that made us fix the errors.

Thanks a lot for your detailed and professional comments, it is very effective in improving the quality of the paper.

Reviewer 3 Report

This paper estimated the cultivated land use efficiency of 342 prefecture-level administrative regions in China from 2003 to 2019 using the stochastic frontier production function, and used spatial autocorrelation analysis and the Gini coefficient decomposition model to explore the spatial agglomeration and spatial disequilibrium of cultivated land use efficiency in China. My comments are as follows.

1. Is there any policy suggestions derived from this paper?

2. This paper used the case study of China. The international implication should be added.

3. Most of the references are from China. The references over the world should be illustrated.

4. The validation of the results should be added.

5. Why apply these methods or models in this paper? The applicability of the methods or models should be explained.

6. The research gap should be mentioned in Section 1 ‘Introduction’.

Author Response

This paper estimated the cultivated land use efficiency of 342 prefecture-level administrative regions in China from 2003 to 2019 using the stochastic frontier production function, and used spatial autocorrelation analysis and the Gini coefficient decomposition model to explore the spatial agglomeration and spatial disequilibrium of cultivated land use efficiency in China.

My comments are as follows.

Q1: Is there any policy suggestions derived from this paper?

R1: Thanks for this comment. We have added more policy suggestions by combining the research findings with the literature. On the one hand, The positive aggregation of cultivated land use efficiency in China has become more pronounced over time. That is, high value areas of cultivated land use efficiency tend to be ad-jacent to other areas of high levels of cultivated land use [1]. Therefore,This should focus on the "spatial linkage" of cultivated land use efficiency, deepening the spatial diffusion effect, reducing the spatial polarization effect, and strictly preventing and controlling cultivated land in the areas with high efficiency of cultivated land utilization. On the other hand, inter-regional differences are the most important source of imbalance in the efficiency of cultivated land use. Therefore, an effective way to reduce the degree of spatial non-equilibrium would be by narrowing the gap of cultivated land use among regions and strengthening the sharing and association of agricultural resources and agricultural technologies in different regions, especially the coordination and development between adjacent cities [2]. Finally, we combined existing studies to partition the results of the measurement of cultivated land use efficiency [3]. For example, HH-type areas can be divided into HH Plain Key Development Zone, which can be used as a national key grain production base and bear important grain production responsibilities.

  1. Wang, Y.; Rao, Y.; Zhu, H. Revealing the Impact of Protected Areas on Land Cover Volatility in China. Land 2022, 11, doi:10.3390/land11081361.
  2. Zang, J.M.; Tang, C.Y.; Wang, Q.X.; Li, K.; LI, L.F. Research on Spatial Imbalance and Influencing Factors of Cultivated Land Use Efficiency in Guangdong Province Based on Super-SBM Model. China Land Science 2021, 35, 64-74, doi:10.11994/zgtdkx.20211008.085358.
  3. Xiong, C.S.; Zhang, Y.L.; Wang, Y.J.; Luan, Q.L.; Liu, X. Multi-function Evaluation and Zoning Control of Cultivated Land in China. China Land Science 2021, 35, 104-114, doi:10.11994/zgtdkx.20210916.155106

Q2: This paper used the case study of China. The international implication should be added.

R2: Thanks for this comment. We have added international implication.

Q3: Most of the references are from China. The references over the world should be illustrated.

R3: Thanks for this comment. We have added more references from around the world.

Q4: The validation of the results should be added.

R4: Thanks for this comment. We have added the verification of the results. We have added the model applicability test and parameter estimation test.

Methods  

Q5: Why apply these methods or models in this paper? The applicability of the methods or models should be explained.

R5: Thanks for this comment. We have inserted some sentences to explain these. The stochastic frontier analysis used in this paper belongs to the parametric method, which separates the inefficiency term from the random error term to ensure the efficiency of the estimated individual, and considers the influence of random error on in-dividual efficiency. At the same time, Dagum Gini coefficient method can solve the problem of decomposition and sample description in regional differences. Therefore, these methods were applied in this study.

Q6: The research gap should be mentioned in Section 1 ‘Introduction’.

R6: Thanks for this comment. We have inserted some sentences to introduce the research gap. On the one hand, existing studies are less concerned with the spatial distribution characteristics of cultivated land use efficiency at the national urban scale; on the other hand, existing studies mostly use DEA or SBM models, which cannot fully consider the effects of stochastic factors and technical inefficiencies. Therefore, the present study fills these research gaps.

Thanks a lot for your detailed and professional comments, it is very effective in improving the quality of the paper.

Round 2

Reviewer 2 Report

Thank you very much for revising your manuscript according to the comments. The revisions were made satisfactorily. 

Just one slight adjustment:

L133-134: "...into the constant price in 2003" should better read "...into constant prices using the base year 2003." Please also explain what kind of deflator was used. 

Reviewer 3 Report

The authors have fully responded to my comments.